# Is there an ex-ante moral hazard on Indonesia's health insurance? An impact analysis on household waste management behavior

**Beta Yulianita Gitaharie** [ID]*, **Rus'an Nasrudin** [ID], **Ayu Putu Arantza Bonita** [ID], **Lovina Aisha Malika Putri** [ID]ᵒ, **Muhammad Abdul Rohman**ᵒ, **Dwini Handayani**

Department of Economics, Faculty of Economics and Business, Universitas Indonesia, Depok, Indonesia

ᵒ These authors contributed equally to this work.
* beta.ylaksono@gmail.com

**Data Availability Statement:** Data is available from https://www.rand.org/well-being/social-and-behavioral-policy/data/FLS/IFLS/ifls4.html.

## Abstract

The presence of ex-ante moral hazard could undermine the potential gain from expanding health insurance coverage in developing nations. To test the proposition, this study utilizes a nationally representative longitudinal survey with Indonesia's health insurance for poor policy in 2014 as the quasi-experimental case study. The country represents developing nations that undergo a massive and rapid expansion of health insurance coverage. The empirical approach combines a matching and difference-in-differences method to obviate potential bias of the selectivity nature of health insurance provision and time-invariant unobserved factors. The findings suggest the presence of ex-ante moral hazard in the form of the less people using trash cans associated with the introduction of the subsidized health insurance premium. The results add empirical findings of a negative side effect of expanding health insurance coverage in developing nations.

## 1 Introduction

The governments of developing nations use various schemes to reach universal health coverage [1]. Among countries' schemes, Indonesia implements subsidized health insurance premium to poor households known as Premiums Assistance Recipient (PBI), launched in 2014 (2) and as an earlier version as *Askeskin* [2]. One of the key motivations was to protect these vulnerable groups from catastrophic health spending upon illness [3–5]. Despite this merit, the scheme is believed to potentially create an ex-ante moral hazard [6–8]. In this study, we revisit the issue by investigating the preventive behavior of lower-income households, in the sense of living hygienic, particularly domestic waste management, after they obtain subsidized health insurance from the GoI (Government of Indonesia). We expand the hypothesis of the effects considering the conventional adverse effect in the form of the increased careless behavior and the improved health awareness because of the increase in healthcare visits.

The presence of moral hazard is an important consideration in developing an optimal health insurance design [9]. Its existence may reduce the effectiveness of health insurance

**Funding:** The authors are grateful for the support of the PUTI Grant by the Directorate of Research and Development, Universitas Indonesia (NKB-2043/UN2.RST/HKP.05.00/2020).

**Competing interests:** This research and publication has no competing interests related to any patents, patent application, or products in development or for market.

provision. Discussion on ex-ante moral hazard has received little subsequent attention in empirical work of health insurance [10]. The existing empirical research in several countries that confirm the presence of ex-ante moral hazard in health insurance are more related to unhealthy lifestyle, such as consuming high-calorie foods [11], smoking and drinking alcohol [12–14], becoming obese [15], doing less physical activities/exercises [14], and spending a large amount of time in sedentary activities ([11] and [16]). Insured households in Ghana are also less likely to use bed nets to prevent malaria [17]. To the best of our knowledge, literature on ex-ante moral hazards that relate health insurance to a clean environment or hygienic living conditions is lacking. The relation is still not unveiled. For that reason, we intend to fill the empirical gap by investigating the impact of subsidized health insurance provision for lower-income households on their waste management behavior in Indonesia.

The empirical approach of this study utilizes a nationally representative longitudinal survey in Indonesia, namely IFLS (Indonesia Family Life Survey) of the Rand Corporation. The data represents the nation's population, and the samples mainly come from the country's major islands, including Sumatra, Java, and Kalimantan islands. We exploit the variation of the household's health insurance between the baseline period of 2007 and the end line period of 2014 as the most suitable recent data that is available for analysis. To the best of our knowledge, there is no other potential data for such purpose. The socioeconomic national survey (SUSE-NAS) has some thematic survey in particular year containing health aspect, such as in 2017. In addition, the Demographic Health Survey/DHS (SDKI) and the Basic Health Research Survey (RISKESDAS) are potential but do not contain our variables of interest. Nevertheless, these datasets are cross-sectional so they are less ideal for our purpose.

The focus of our study is to investigate whether there is a different behavior of waste management between the insured and uninsured lower-income households. We exercise differences in pre-treatment level between insured and uninsured to directly compare both groups' characteristics and perform propensity score matching combined with a difference-in-difference procedure. These approaches are used to ensure that our impact estimate is a result of the subsidized insurance and not due to the common behavior of a specific group. We also run a heterogeneous analysis and coefficient stability for robustness.

This study aims to test the causal inference between the provision of subsidized health insurance for lower-income households and how these households manage their waste. This study contributes to empirical evidence by examining whether ex-ante moral hazard behavior exists among Indonesia's poor and vulnerable groups. There are two plausible competing effects in which subsidized health insurance could affect individuals—adverse and positive effects. The adverse effect arises when the behavior of the health insurance holder does not take care of his health because he knows the insurance will reimburse his medical expenses if he becomes ill. The positive effect works through the availability of health promotion information and greater contact with medical professionals. Health insurance can be perceived as an important factor in increasing preventive care and possibly improving health behavior [18].

Based on our estimates, we find evidence of ex-ante moral hazard in the subsidized health insurance recipients on how they manage their household waste. The insured households are less likely to use trash can to dispose their domestic waste than the uninsured. The effects vary by demography and geography, suggesting some important implications for focusing the mitigation efforts. The finding of this study is expected to help the Indonesian government to design a better intervention concerning adverse side effect of health insurance subsidy.

The following section describes the plausible competing effects on recipients of subsidized health insurance. Section 3 presents the existing condition of household waste management and health insurance in Indonesia. Section 4 depicts the empirical strategy and data employed. Section 5 displays the empirical result and discussion. Lastly, Section 6 concludes the paper.

## 2 The competing effects of subsidized health insurance: Ex-ante moral hazard vs. the improved awareness of hygiene

Theoretically, there are two plausible effects in the discussion of health insurance: one positive and the other one is adverse. One positive effect of owning health insurance is that the owner would have more opportunities to communicate and interact with medical professionals. If healthcare professionals advise their patients about health risks, the increased interaction and the patients' improved knowledge could lead to better health behavior [19, 20]. A study in Saudi Arabia finds that health insurance holders are more likely to seek medical attention—the effects are higher amongst non-Saudi nationals compared to Saudi Arabian citizens—and are encouraged to get specialized medical exams for high cholesterol, diabetes, and hypertension [21]. Meanwhile, a study in China explains that public health insurance (PHI) has heterogenous effects depending on age and income [22]. The findings show that the effect of PHI is stronger for both middle-aged and elderly. When it comes to the lower income group, PHI significantly increases the probability of having no chronic diseases, self-reporting good health, and good life satisfaction (mental health). The low-income group also tends to have a more significant increase in health care utilization and is more likely to feel relieved of a medical financial burden. In the rural area of Uganda, the presence of voluntary community-based health insurance increases the likelihood of utilizing a mosquito net and getting dewormed by 26% and 18%, respectively [23]. They hypothesize that health insurance has an impact through three different channels: the financial protection channel, the use of healthcare services, and the dissemination of information and social learning.

Even though health insurance is meant to protect the holders from health and financial risks, its ownership is not always followed by healthy living behavior. There are still insurance participants who behave recklessly, leading to an ex-ante moral hazard and resulting in an adverse outcome. To mention some of the ex-ante moral hazards in health insurance, as described earlier in Section 1 of this paper, are the findings of [11–17] which is in contrast to [23].

Mixed results on health behavior are also found in the case of the Affordable Care Act (ACA) study conducted by [18, 19, 24]. A study by [18] finds that the after-one-year ACA expansion increases insurance coverage, access, and the use of certain forms of preventive care; but they do not find evidence of ex-ante moral hazard. Meanwhile, the three-year ACA expansion also increases preventive care utilization but increases risky behavior, hence, ex-ante moral hazard occurs [24]. For the five-year ACA expansion, as in [18] finds the expansion increases utilization of certain forms of preventive care, reduces heavy drinking and smoking, and increases the probability of exercise [19].

Many studies, some as mentioned above, focus more on the effect of health insurance on lifestyle, for example, consuming high-calorie foods, smoking and drinking alcohol, doing less physical activities/exercises, spending a large amount of time in sedentary activities, and becoming obese. As far as we can tell, there is no literature yet on ex-ante moral hazards that link health insurance to hygienic living conditions. Poor living conditions which are commonly found in developing countries, including improper domestic waste management issues, are associated with health problems. This paper questions how would the causal effect of subsidized health insurance on domestic waste management in Indonesian households, specifically in the lower-income group.

## 3 National health insurance and household waste management: The Indonesia context

### 3.1 Indonesia national health insurance

Since its establishment during the colonial period, Indonesia's health insurance program has evolved. One of the milestones is initiating universal health coverage in 2004 through

Askeskin, which provides health insurance for the poor [25]. A study conducted by [2] shows that the Askeskin program has targeted the poor and the vulnerable, despite the non-trivial leakages to the non-poor. Askeskin has led to an increase in outpatient healthcare utilization among the poor. [26] extends the study of [2] by breaking down the samples into subgroups and investigates the impact of Askeskin on both outpatient and inpatient healthcare. The author finds that the program had a larger effect on the use of outpatient healthcare by females than males, but inversely on inpatient healthcare.

Askeskin then transformed to Jamkesda (*Jaminan Kesehatan Daerah*) in 2005 and Jamkesmas (*Jaminan Kesehatan Masyarakat*) in 2008 [27]. Both were dedicated for the poor and the vulnerable. In term of their sources of financing, Jamkesmas is financed by the State Budget (APBN-*Anggaran Pendapatan Belanja Negara*), whilst Jamkesda by the Regional/Local Government Budget (APBD-*Anggaran Pendapatan Belanja Daerah*) to cover the shortfall in funding from Jamkesmas in the region.

Intending to provide comprehensive, fair, and equal health coverage to the entire population, including the poor and vulnerable [28], the GoI launched JKN (*Jaminan Kesehatan Nasional* or the National Health Insurance) in 2014. Not only for free treatment, but JKN also provides the opportunity for people to be healthy by reducing risk and screening those at risk [29]. To that end, the BPJS (*Badan Penyelenggara Jaminan Sosial* or Social Health Insurance Administration Body) has curative and preventive action programs as the managing entity for JKN. JKN has covered 82.3% of Indonesia's population as of March 2021, but it still has a long way to meet the 98% target by 2024.

To protect the poor and vulnerable, the GoI offers PBI (*Penerima Bantuan Iuran*-health insurance premium assistance beneficiary). The targeted group and the amount of contribution are both determined by the government—through the Ministry of Social Affairs Regulation No. 21 of 2019 and the Indonesian Presidential Regulation No. 64 of 2020, respectively. The criteria of the targeted include: (i) not having a source of livelihood and or having but do not meet basic needs, (ii) having expenses only to meet basic needs, (iii) being unable to afford for seeking medical attention, (iv) unable to afford for buying clothes once a year for household members, (v) able to send their children to junior high school, (vi) having the walls of the house made of bamboo/wood/wall in poor condition/low quality, (vii) having the floor of the house made of soil or wood/cement/ceramic with poor condition, (viii) having the roof of the house made of palm fiber/*rumbia* or tile/tin roof/asbestos with poor condition, (ix) having house lighting not from electricity or electricity without a meter, (x) having small house floor area, less than eight meters-squares /person; and (xi) having drinking water sources come from wells or unprotected springs/river water/rainwater/other. To these targeted groups, the health insurance contributions for PBI participants are paid in full by the government with 42,000 IDR per month per eligible recipient.

With the implementation of JKN, Jamkesda and Jamkesmas participants automatically become PBI participants. The PBI participation has increased by 40% since the scheme was first launched in 2014 to 2019. Fig 1 indicates that, as of July 2019, 83% of the total population is insured, and 60% of which is that of PBI.

The other beneficiary group is the non-PBI which covers (i) wage workers and family members, (ii) non-wage workers and family members, (iii) non-workers and family members. The non-PBI participation has also grown by 132% for 2014–2019. Non-PBI participants, on the other hand, must pay contributions as depicted in Table 1.

Despite its growing participants, BPJS has faced main challenges-the number of participants remains below the target, there is still inequality in access to health services, and there are issues related to health service financing [30]. Since its first establishment, BPJS has still run deficits in its financial statement. It is likely because participant contribution is lower than the actuarial calculation value (see Fig 2). In addition, it is recorded that 25,326 companies

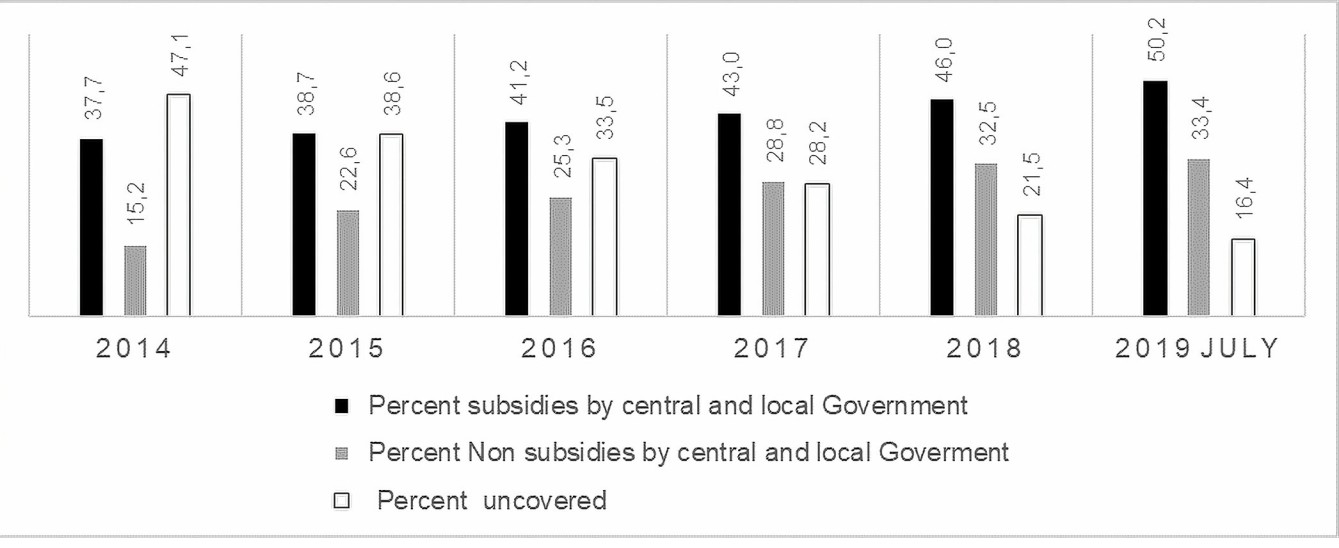

**Fig 1. National Health Insurance Indonesia (number of people).** Source: Social Security Agency for Health.

manipulate employee data potentially cause revenue losses of IDR 6.19 billion [31]. Fig 3 shows that the total expenditure for health insurance has exceeded its contribution revenue since the JKN program first launched in 2014, except in 2016 and 2019.

Table 2 shows that JKN utilization among PBI participants remains far below that of non-PBI participants. It indicates that the GoI needs to focus on increasing health literacy among the poor. People with poor health literacy do not look for medical services when needed. Despite health coverage [32], they habitually visit health care facilities only when their diseases have reached an advanced stage. Being well informed is essential to promote positive results in healthier lives.

## 3.2 Household waste management in Indonesia and related diseases

As in other developing countries, household waste management remains a major concern in Indonesia. The country is ranked fourth in the world in generating municipal solid waste

**Table 1. Premium contribution by non-PBI members according to the JKN scheme.**

| No | Members/participants | Contributors to premium payments | The amount of contributions |
|---|---|---|---|
| 1 | Government civil servants, members of the Indonesian National Armed Forces, Indonesian National Police | Participants, Central governments, and Local governments | 5% of salary or wages per month (4% is paid by employers and 1% by participants) |
| 2 | Salaried workers (*Pekerja Penerima Upah*) in the Central Government Enterprises (BUMN) or in the Local Government Enterprises (BUMD) as well as in the private sector | Participants/employees and employers | 5% of salary or wages per month (4% is paid by employers and 1% by participants) |
| 3 | Nonsalaried workers (*Pekerja Bukan Penerima Upah*) and non-employees (*Bukan Pekerja*) | Participants | (a) IDR 42,000 per member per month with benefits services in the 3rd class ward. Participants only pay IDR 35,000 for the contribution, and the GoI subsidizes IDR 7,000. |
| | | | (b) IDR 100,000 per member per month with benefits services in the 2nd class ward. |
| | | | (c) IDR 150,000 per member per month with benefits services in 1st class ward. |

Source: Presidential Regulation 64 of 2020

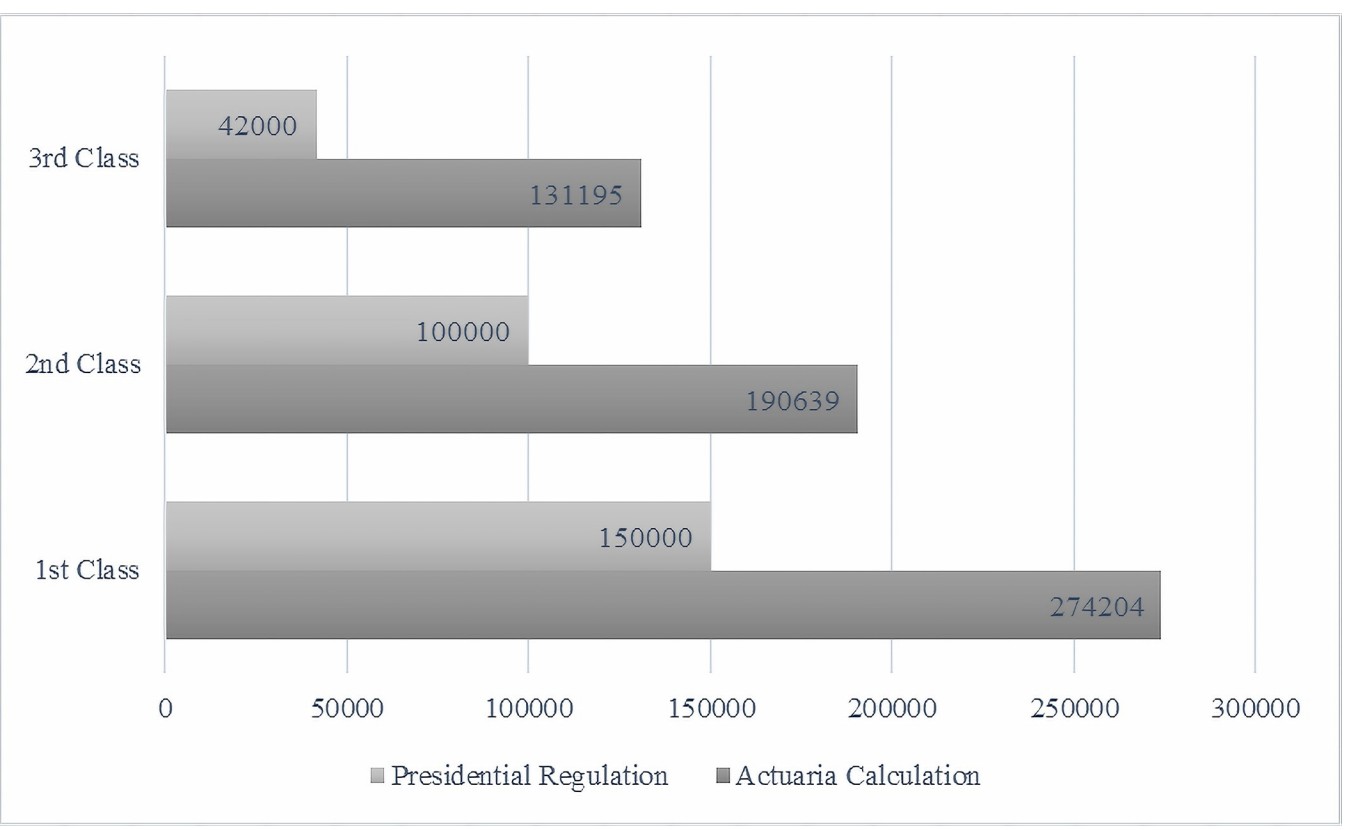

**Fig 2. Insurance premium comparison: Actuarial estimation and presidential regulation (in IDR).** Source: The Audit Board of The Republic of Indonesia.

(3.55%) after the US (4.40%), India (18.05%), and China (18.75%) [33]. Indeed, households in Indonesia generate waste the most (45.4%) [34]. The four largest types of waste are food waste (40.1%), plastic (17.2%), wood/leaves/twigs (14.2%), and paper/cardboard (11.9%) [34].

Most households do not manage their waste properly. Regardless where they reside, households burn (rural: 68.2%; urban: 34.1%; Indonesia: 49.5%), dispose their garbage in the river/sewer/sea (rural: 11.3%; urban: 4.9%; Indonesia: 7.8%), and anywhere improperly (rural: 10.1%; urban: 2.5%; Indonesia: 5.9%) [35]. Particularly, among the lower income group (quintiles 1 and 2) 55.95% of them do not manage waste based on our calculation with IFLS 2014.

Households need to be aware that waste evolves in time. Waste material composition becomes more complex and frequently contains toxins and harmful elements [36]. Waste sorting is, therefore, urgent to do. A survey by KataData Insight Center [37] indicates that 50.8% of households interviewed do not separate nor sort their domestic waste. They argue that—they do not want to bother with waste sorting (79%), waste will eventually be mixed at the temporary/final disposal sites (17%), it is useless to sort waste (3%), and other (1%).

The home environment is one crucial determinant of resident health [6]. People with lower incomes are more likely to have poor health [4, 38]. It is globally indicated that more than 8,000 people die every day from poor sanitation and hygiene conditions [39]. Based on BPS data in 2019, the Indonesia lower-income group or people living in poverty covers 9.41% of the population [40]. As for rural-urban division, 43.3% of population are living in rural and 56.7% in urban areas; and 13.86% living in an unclean environment [41]. The untreated waste —44% of 252 square meters of the estimated household waste production per day [42]–may become vectors of diseases, including malaria and dengue [36]. The number of dengue cases

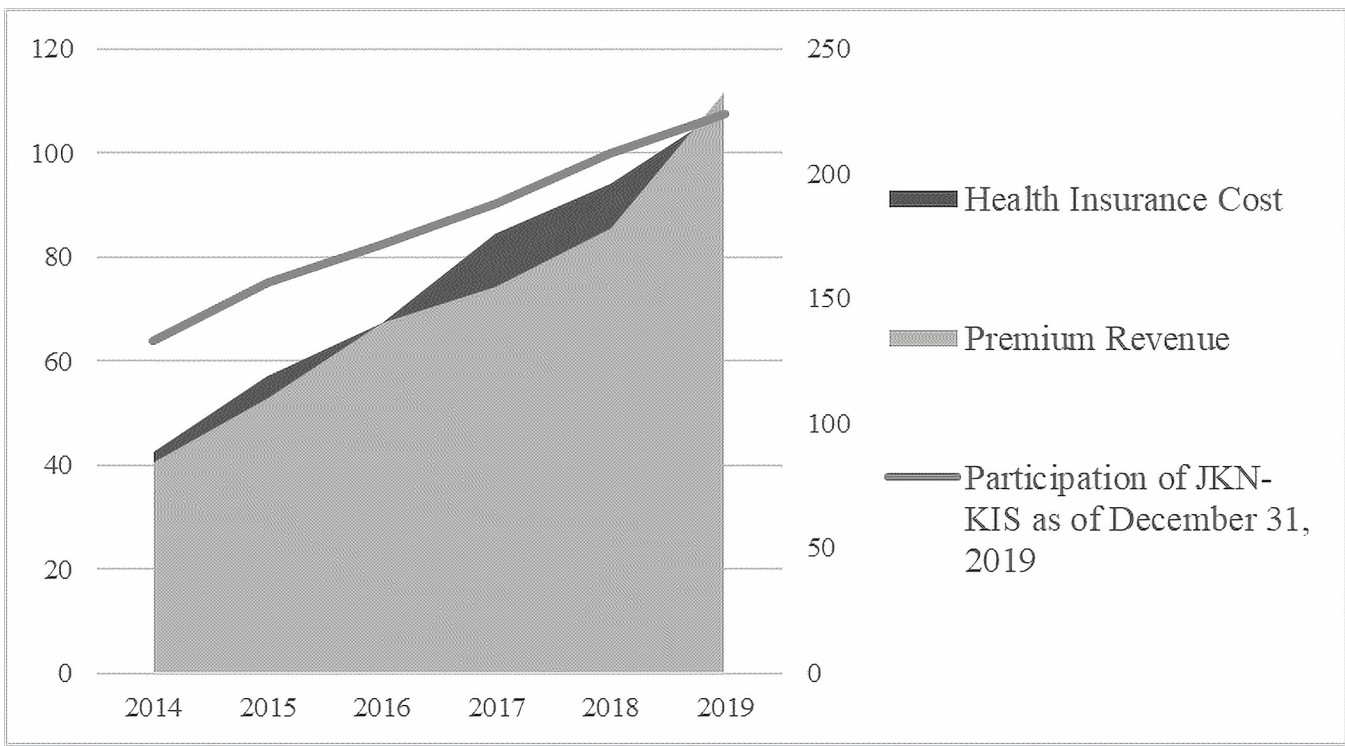

**Fig 3. Participation of JKN-KIS as of December 31, 2019 (IDR million), health insurance cost (IDR trillion), and premium revenue (IDR trillion).**
Source: Social Security Agency for Health.

in Indonesia has increased from 68,410 in 2017 to 137,760 cases in 2020 [43]. In terms of home environment-related diseases [6–8], specifically, the subsidized health insurance recipient households suffer from chronic diseases, primarily hypertension, digestive disease, and arthritis/rheumatism (see S7 Table in S1 Appendix), and acute diseases—headache, runny nose, cough, stomach ache, skin infection, and diarrhoea (see S8 Table in S1 Appendix). As a

**Table 2. Subsidized insurance utilization rate (per mil).**

| Panel A–official data | | | | | |
|---|---|---|---|---|---|
| Health Insurance Beneficiary Group | Year | | | | |
| | **2014** | **2015** | **2016** | **2017** | **2018** |
| PBI | 36.3 | 35.3 | 38.3 | 43.9 | 45.2 |
| Non-PBI | 920.6 | 772.9 | 827.1 | 930.9 | 999.9 |
| Panel B–IFLS Data (Number of households [%]) | | | | | |
| Health Insurance Beneficiary Group | Year | | Total | | |
| | **2007** | **2014** | | | |
| Non-subsidized insurance | 9,034 [84.62] | 5,677 [53.18] | 14,711 | | |
| Subsidized insurance | 1,642 [15.38] | 4,999 [46.82] | 6,641 | | |
| Total | 10,676 [100.00] | 10,676 [100.00] | 21,352 | | |

Source: Fiscal Office, Ministry of Finance, 2020. Authors' calculation. The figures are only balanced panel households, excluding attrition and newly sampled households.

preventive action, improving the home environment, sanitation, and hygiene could reduce the incidence of sanitation-related diseases.

Waste management in Indonesia is a reasonably complex issue. Indonesia faces both demand and supply constraints. There is an increasing volume and production of waste from the demand side as the population grows and urbanization rapidly expands. Slums in urban areas have also doubled—rising from 4.09 percent in 2018 to 9.04 percent in 2019 [42]. The public lacks knowledge in proper waste management. On the supply side, there is inadequate provision of waste facilities, equipment, and technology. Only 27.15% of household waste in Indonesia was collected by the garbage collector in 2017 [42] and served by 3,873 garbage trucks across 34 big cities in the county. The availability of garbage handcarts that usually serve door-to-door and temporary waste storage (TPS-*Tempat Pembuangan Sementara*) has declined by 20% and 4.84%, respectively, in 2019 [42].

## 4 Data and empirical strategy

### 4.1 Data

The data for this study comes from the Indonesia Family Life Survey (IFLS). IFLS is a comprehensive and nationally representative longitudinal survey conducted by Rand Corporation in Indonesia. The surveys consist of five waves in which we utilize the last two waves of IFLS 4 (the year 2007) and IFLS 5 (the year 2014) for our purpose of the study. The ethical review and clearance follow the IFLS ethical clearance. The survey covers socio-economic and health variables that allow us to identify household ownership changes toward health insurance and associate them with the reported actions of responsible health and environmental behavior representing the ex-ante moral hazard measures. The behavioral measures include toilet ownership, where the household drains its sewage, how the household disposes of its garbage, and whether the household participates in health funds by the community.

### 4.2 Empirical strategy

The main objective of our study is to examine the impact of lower income group household health insurance ($SI_{ht}$) ownerships status on ex-ante moral hazard behavior, measured by household's waste management behavior measures ($Y_{ht}$). We use Propensity Score Matching (PSM) (34) concerning the subsidized insurance ownership selection issue in the pre-estimation step. Subsidized insurance in Indonesia is a household targeted program with the eligibility criteria of living under poverty. The matching strategy aims to create a comparable control group for those who own subsidized insurance. Therefore, our impact estimate will better reflect the implication of the subsidized insurance on waste management behavior and is not because of the different characteristics between the two groups. A similar approach of matching for analyzing subsidized insurance is a norm in the literature. See for example [17] or [2].

Some anti-poverty programs in Indonesia have imperfect targeting performance [44, 45], with PBI or Askeskin as no exception. There are cases of inclusion (non-eligible households receiving program) and exclusion error (eligible households but not receiving the program). Fig 4 indicates these targeting errors. However, the imperfect targeting allows us to construct a counterfactual group using similar households who did not receive the program at the baseline period ($SI_{i,2007}$) with the following matching strategy:

$$P(SI_{i,2007} = 1 | Z_{h,2007}) = F(\hat{\theta} Z_{h,2007}) \tag{1}$$

in which $Z_{h,2007}$ represent a set of household characteristics that the Indonesian government uses as the eligibility criteria for subsidized insurance targeting. It includes the age of the

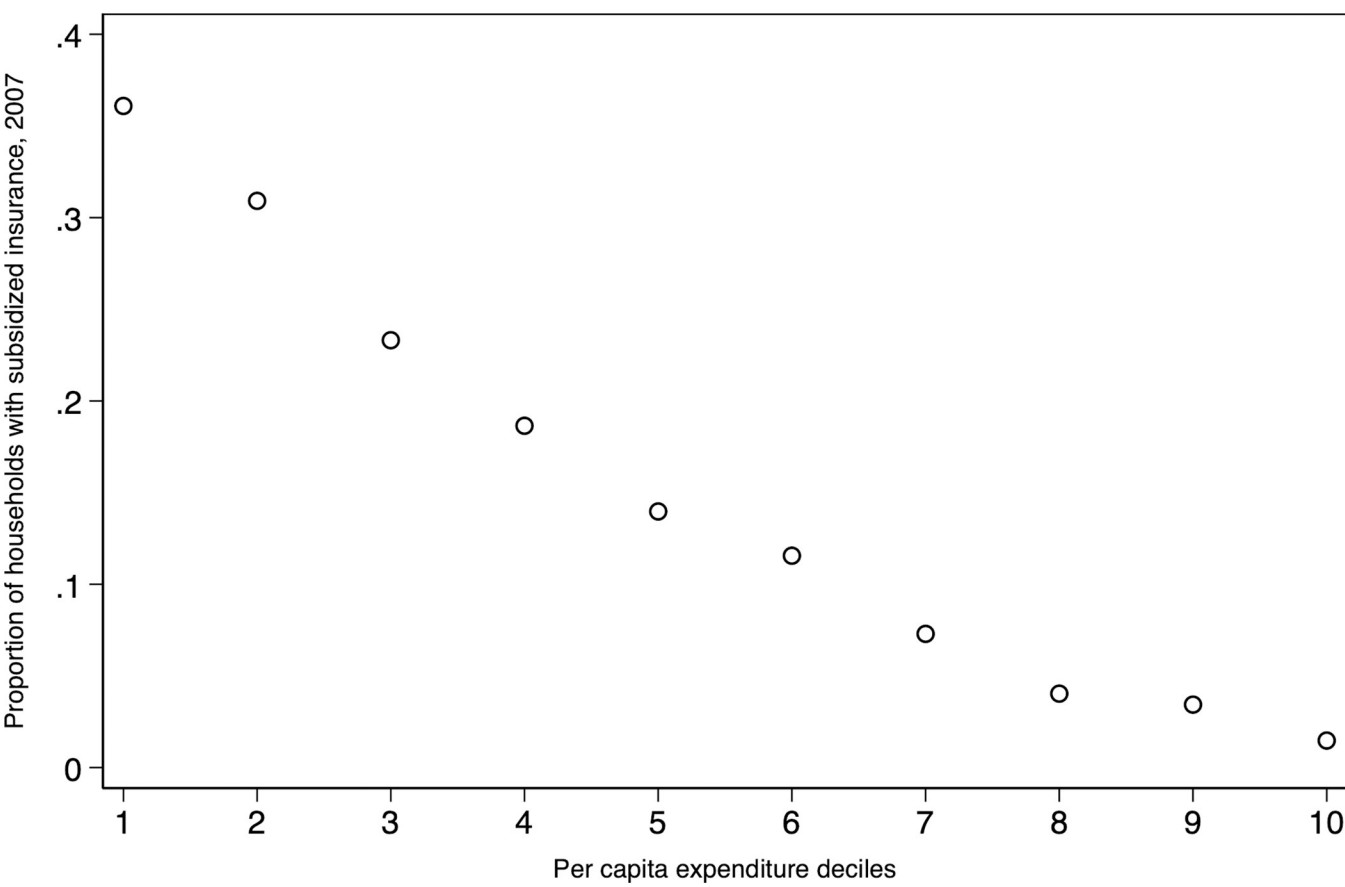

**Fig 4. Subsidized insurance's take up-rate by expenditure decile, 2007.** Source: Author's calculation with IFLS 2007.

household head, per capita expenditure, household size, gender of household head, marital status of household head, house conditions (has a bamboo wall, soil floor, has the vehicle, has an electronic asset, has electricity access, and ownership of toilet). The estimation for $F(\hat{\theta}Z_{h,2007})$ utilizes limited dependent variable estimator of a *Probit* model.

Subsequently, if both treated and comparison group resulting from matching strategy has similar (parallel) growth pattern of their potential ex-ante moral hazard behavior in the absence of subsidized insurance, we estimate Eq (2) to the observations that are in the common support of the propensity score matching obtained from Eq (1) with a double difference framework. For household $h$ and time $t$, the main specification of the canonical difference-in-differences estimator to identify the impact of $SI_{ht}$ ownership on waste management outcome ($Y_{ht}$) between pre and post period indicated by $POST_t$ with the parameter of interest $\beta_3$ is the following:

$$Y_{ht} = \alpha + \beta_1 SI_h + \beta_2 POST_t + \beta_3 SI_h \times POST_t + X_{ht}\gamma + \eta_{rt} + \varepsilon_{ht} \qquad (2)$$

The point of estimate of $\beta_3$ captures the difference-in-differences in the outcome variables concerning waste management between households that have SI and similar ones that do not have SI across times before and after SI implementation. The nature of endogeneity of measuring the impact of SI ownership on ex-ante moral hazard could originate from unobserved time variant (such as degree of risk aversion) and time invariant factors (such as expansion of supply-side health-related public infrastructure), apart from the selectivity problem. Therefore, to

control for time-varying observable confounding factors, we include a set of household-level covariates of $X_{ht}$ that covers family years of education, degree of risk aversion, the availability of waste management supply at the district level, population density, and rural-urban dummy in addition to unbalanced covariates explained in Section 5.2. Last, we also include island-year dummies of $\eta_{rt}$ to limit heterogeneity bias originated from macro aggregate shocks and the varying development level in the Indonesian archipelago. We also perform the coefficient stability test using the framework developed by [46]. The test uses assumptions of the maximum R-squared equals 1.3 times the existing model with complete control variables and "the proportionality assumption" of 0.545.

We also perform heterogeneous analysis using subgroups which include gender, education, income quintiles, expenditure per capita, location—both rural-urban and region, and waste service to observe the group that drives our main result.

## 5 Result and discussion

### 5.1 Defining treatment and control group

To identify the treatment and control groups, in IFLS, there are four categories of individuals based on their status of subsidized insurance ownership: always treated, never treated, newly treated, and untreated. Between 2007 and 2014, always treated individuals were those who identified held subsidized insurance in both periods. In 2007 subsidized insurance is elicited by *Askeskin* as the only available survey question item, whereas in 2014 it expanded into a wider type of subsidized insurance including *Jamkesda*, *Jamkesmas*, *Jamkessos*, *Jampersal and JKN* as the new terminologies and policies. Those who never had subsidized insurance in baseline or end line period are never treated. Newly treated did not receive *Askeskin* in 2007 but then held any subsidized insurance in 2014. Last, the untreated had *Askeskin* in 2007 but no longer had any subsidized insurance in 2014. We only use always and never treated as treatment and control group to avoid bias from subjects with switching status as they created negative weighting issues in the estimation [47]. Effectively, we use only 1,642 households as the treatment group and 5,677 households as the control group among a total of 10,676 balanced panel households. The flow of subsidized insurance status changes between 2007 and 2014 in IFLS is provided in Fig 5 and Table 2 Panel B. The descriptive statistics is presented in S9 Table in S1 Appendix. They also show that the expansion of subsidized insurance has been massive from 2007 to 2014.

### 5.2 Pre estimation

Table 3–Panel A presents the pre-treatment balance test results for the covariates to determine the probability of getting subsidized insurance. The result shows that most of the variables have a statistically significant difference between the mean for the control and treatment groups. These characteristics gap between the two groups indicates that subsidized insurance ownership is a selective event.

Furthermore, when we estimate Eq (1) the statistical significances are also picked up by the same variables in the t-test mentioned above. As presented in Table 3 Panel C, the result indicates that households with good economic conditions are less likely to participate in subsidized insurance. A household with higher per capita expenditure is less likely to be insured. Moreover, households who own electronic assets and vehicles are 1.8% and 4.5%, respectively, less likely to be insured. On the other side, households who live in a house with a wall made of bamboo and have no toilet at home are 6,1% and 5,2%, respectively, more likely to have participated in subsidized insurance.

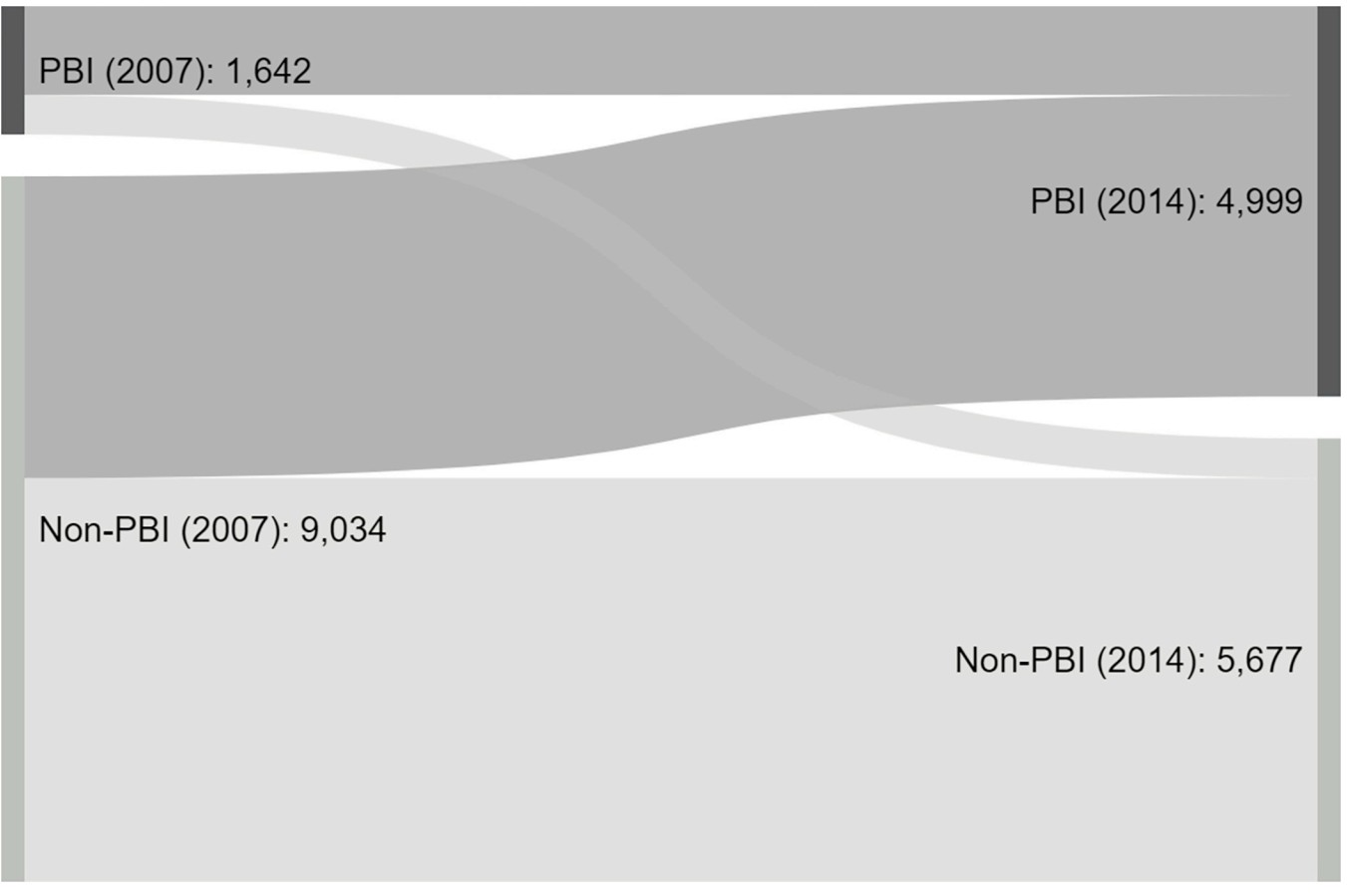

**Fig 5. Shifting of subsidized insurances beneficiaries during 2007–2014.** Source: Author's calculation using IFLS Wave 4 & 5.

To limit the potential bias from the selection mechanism, we perform PSM and calculate the propensity score of each individual. We obtained 35 off-support individuals using the matching method of an Epanechnikov kernel with a 0.06 bandwidth. The estimate suggests using 1,642 treatment individuals and 5,642 control individuals after excluding 35 samples from the untreated group. However, the balancing test in Panel B of Table 3 suggest that not all of the selection variables are balanced. Only household head gender is balanced between the treatment and control group but not for the rest. To anticipate bias associated with the imperfection in the matching step, we add all of the unbalanced covariates in the main estimates. The distribution of propensity score by treatment status and on-support is provided in Fig 6.

## 5.3 Main estimate

This study hypothesizes that participating in subsidized health insurance could lead to two possible effects in the form of a hygienic home environment. One is a more responsible behavior toward better hygiene and environmentally responsible behavior resulting from the indirect health educational-process obtained from health services. Second, the opposite direction originated from incentive effect in which insurance holders tend to be careless when the adverse health risk is covered.

The main results of estimating Eq (2) only for on-support samples are presented in Table 4 with full covariates. The stepwise estimations by a different specification of covariates are

**Table 3. The propensity score matching estimate.**

| Panel A- The pre-treatment Balancing Tests between Sample with and Without Subsidized Insurance | | | | | | |
|---|---|---|---|---|---|---|
| Covariates | Control | Treatment | Mean Control | Mean Treatment | Diff. | P-value |
| Households head Age | 9,704 | 1,697 | 42.875 | 46.934 | -4.059 | 0.000 |
| Per Capita Expenditure (million Rupiah per month) | 9,485 | 1,687 | 0.667 | 0.381 | 0.286 | 0.000 |
| Household Size | 9,704 | 1,697 | 3.478 | 3.984 | -0.506 | 0.000 |
| HH Head is Male | 9,704 | 1,697 | 0.814 | 0.803 | 0.011 | 0.311 |
| HH Head is Married | 9,704 | 1,697 | 0.783 | 0.802 | -0.019 | 0.071 |
| Wall Bamboo = 1 | 9,704 | 1,697 | 0.055 | 0.134 | -0.079 | 0.000 |
| Floor Not Soil = 1 | 9,704 | 1,697 | 0.931 | 0.865 | 0.067 | 0.000 |
| HH has Vehicle = 1 | 9,704 | 1,697 | 0.581 | 0.456 | 0.126 | 0.000 |
| HH has Electronic Asset = 1 | 9,704 | 1,697 | 0.882 | 0.801 | 0.080 | 0.000 |
| HH has electricity = 1 | 9,704 | 1,697 | 0.967 | 0.932 | 0.035 | 0.000 |
| No Toilet = 1 | 9,704 | 1,697 | 0.111 | 0.228 | -0.117 | 0.000 |

| Panel B- The post-treatment Balancing Tests between Sample with and Without Subsidized Insurance | | | | | | |
|---|---|---|---|---|---|---|
| Covariates | Control | Treatment | Mean Control | Mean Treatment | Diff. | P-value |
| HH Head Age | 8,938 | 1,566 | 42.602 | 46.435 | -3.832 | 0.000 |
| Per Capita Expenditure (million Rupiah per month) | 8,938 | 1,566 | 0.645 | 0.374 | 0.272 | 0.000 |
| Household Size | 8,938 | 1,566 | 3.493 | 4.019 | -0.527 | 0.000 |
| HH Head is Male | 8,938 | 1,566 | 0.814 | 0.811 | 0.003 | 0.820 |
| HH Head is Married | 8,938 | 1,566 | 0.786 | 0.809 | -0.022 | 0.044 |
| Wall Bamboo = 1 | 8,938 | 1,566 | 0.056 | 0.134 | -0.078 | 0.000 |
| Floor Not Soil = 1 | 8,938 | 1,566 | 0.929 | 0.860 | 0.070 | 0.000 |
| HH has Vehicle = 1 | 8,938 | 1,566 | 0.570 | 0.440 | 0.131 | 0.000 |
| HH has Electronic Asset = 1 | 8,938 | 1,566 | 0.878 | 0.793 | 0.085 | 0.000 |
| HH has electricity = 1 | 8,938 | 1,566 | 0.965 | 0.926 | 0.038 | 0.000 |
| No toilet = 1 | 8,938 | 1,566 | 0.115 | 0.235 | -0.120 | 0.000 |

| Panel C-Selection into Subsidized Insurance Regression | | |
|---|---|---|
| Dependent variable: Subsidized Insurance = 1 | | |
| Covariates | Coefficients | Standard errors |
| HH Head Age | 0.001*** | (0.000) |
| Per Capita Expenditure | -0.014*** | (0.002) |
| Household Size | 0.013*** | (0.002) |
| HH Head is Male | -0.015 | (0.012) |
| HH Head is Married | -0.001 | (0.012) |
| Wall is Bamboo | 0.061*** | (0.012) |
| Floor is Not Soil | -0.021* | (0.012) |
| HH has Vehicle | -0.045*** | (0.007) |
| HH has Electronic Asset | -0.018* | (0.011) |
| HH has electricity | -0.015 | (0.017) |
| Has no toilet | 0.052*** | (0.009) |
| N | 11,172 | |

Notes: robust clustered standard error in parentheses with *, **, and *** denotes statistical significance at 10%, 5%, and 1%. Coefficients presented are marginal effects at the mean of Probit regression.

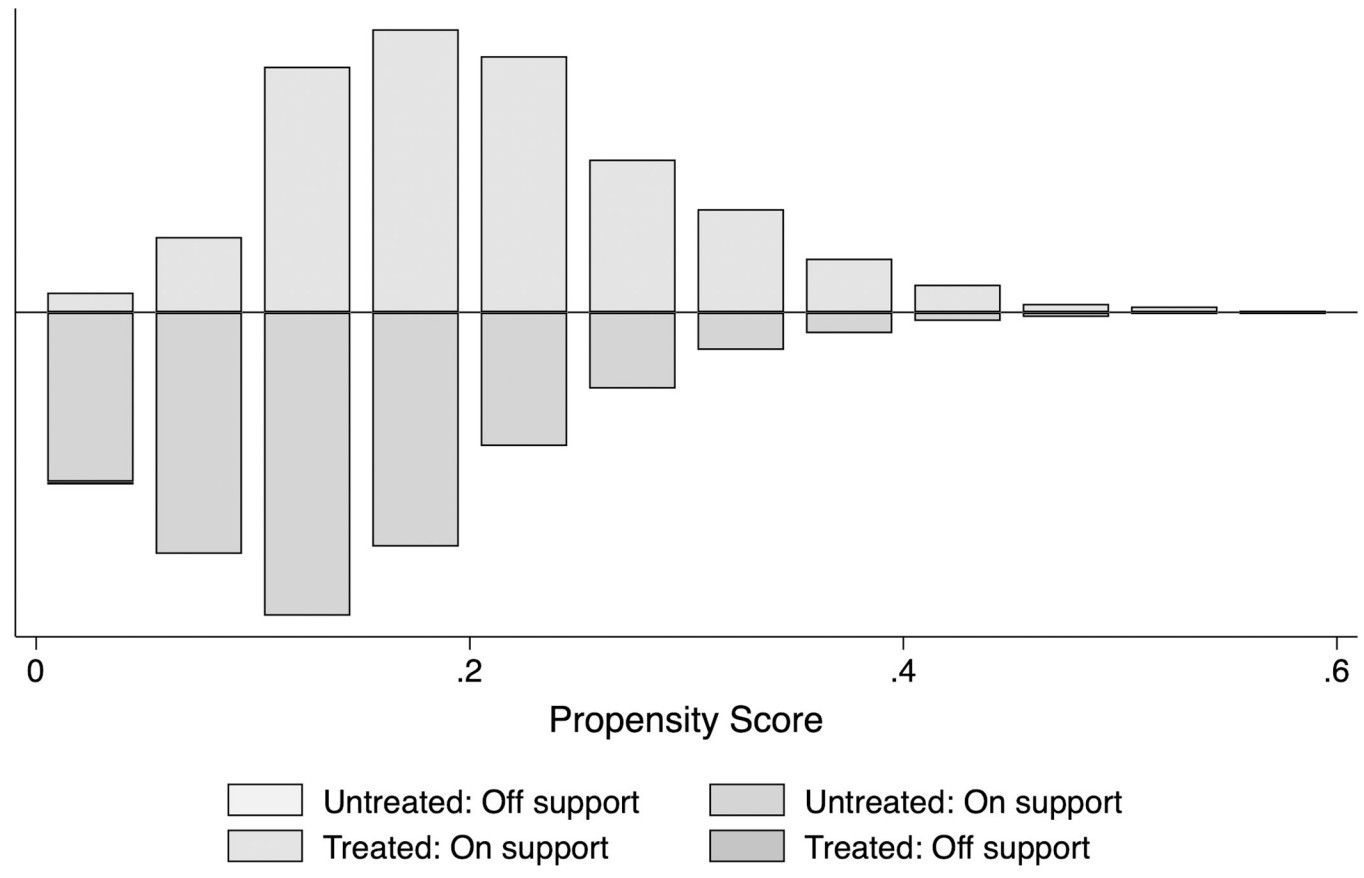

**Fig 6. The propensity score graph.** Source: Author's calculation using IFLS 2007.

shown in S10 Table in the S1 Appendix section. We employ difference-in-differences regression to estimate an ex-ante moral hazard in subsidized health insurance provision on how households manage their waste. The coefficient of interest is the interaction between subsidized insurance and post-period. We found evidence of ex-ante moral hazard concerning the behavior of disposed waste in the trash can, but we see no impact for the other two types of outcomes. As shown in column (1) of Table 4, it is implying that 4 out of 100 insured households are less likely to dispose of waste in the trash can after receiving the treatment compared to their counterfactual.

The ex-ante moral hazard is typical behavior of an insured who does not mind about his/her health and tend to avoid preventive care due to his/her confidence that the medical expenses will be covered by the insurance company. Having insured does not necessarily improve subsidized insurance recipients' knowledge of living a hygienic lifestyle. It contrasts to what is found in previous studies, i.e. [18] and [48].

To this adverse effect of subsidized health insurance provision, we suppose there is a low awareness of proper domestic waste disposal from the demand side, which may be due to a lack of knowledge and habit of doing so [49]. The subsidized insurance recipients may not be aware of the danger of unproper waste disposal since most subsidized insurance recipients have a low educational background. Solid waste harms human health. Household waste, in general, has contaminated water and soil in 21% and 2.7% of Indonesian villages, respectively. It has been reported that 28% of households do not have improved drinking water. In 2016,

**Table 4. Main estimate.**

| | (1) | (2) | (3) |
|---|---|---|---|
| | **Disposed in the Trash Can** | **Burned** | **Disposed in Land, Water, and Yard** |
| Subsidized Insurance = 1 | 0.011 | -0.053*** | 0.055*** |
| | (0.013) | (0.017) | (0.015) |
| POST = 1 | -0.003 | 0.066*** | -0.061*** |
| | (0.015) | (0.021) | (0.020) |
| Subsidized Insurance X POST | -0.039** | 0.038 | 0.006 |
| | (0.018) | (0.024) | (0.021) |
| Years of Education | 0.007*** | -0.002 | -0.008*** |
| | (0.001) | (0.001) | (0.001) |
| Degree of Risk Preference = 1 | -0.003 | 0.010 | -0.006 |
| | (0.008) | (0.012) | (0.011) |
| Degree of Risk Preference = 2 | -0.017 | 0.004 | 0.015 |
| | (0.013) | (0.019) | (0.017) |
| Degree of Risk Preference = 3 | -0.020* | 0.019 | 0.001 |
| | (0.012) | (0.017) | (0.016) |
| Degree of Risk Preference = 4 | -0.007 | -0.005 | 0.013 |
| | (0.009) | (0.013) | (0.012) |
| Cleaning Routine in Community = 1 | -0.000 | -0.006 | 0.001 |
| | (0.007) | (0.010) | (0.009) |
| System of drain/sewage channel = 1 | 0.097*** | -0.064*** | -0.028*** |
| | (0.007) | (0.010) | (0.009) |
| Proportion of Garbage Collector in District Level | 0.470*** | -0.601*** | -0.269*** |
| | (0.015) | (0.026) | (0.027) |
| Log of Population Density (People/Km2) | -0.016*** | 0.035*** | -0.015*** |
| | (0.003) | (0.006) | (0.005) |
| Urban = 1 | 0.147*** | -0.101*** | -0.045*** |
| | (0.008) | (0.011) | (0.010) |
| Island = Java Island | 0.035*** | -0.101*** | 0.070*** |
| | (0.009) | (0.014) | (0.012) |
| Island = Bali, NTT, NTB, Island | 0.029** | -0.186*** | 0.185*** |
| | (0.012) | (0.017) | (0.016) |
| Island = Kalimantan Island | 0.017 | -0.117*** | 0.124*** |
| | (0.014) | (0.022) | (0.019) |
| Island = Sulawesi Island | 0.027* | -0.163*** | 0.143*** |
| | (0.016) | (0.024) | (0.022) |
| HH Head Age | -0.000 | -0.000 | 0.000 |
| | (0.000) | (0.000) | (0.000) |
| Per Capita Expenditure | 0.002*** | -0.002*** | -0.001 |
| | (0.000) | (0.001) | (0.001) |
| Household Size | 0.001 | 0.003 | -0.004 |
| | (0.002) | (0.003) | (0.003) |
| HH Head is Male | 0.002 | -0.017 | 0.010 |
| | (0.011) | (0.016) | (0.015) |
| HH Head is Married | -0.026** | 0.012 | 0.029* |
| | (0.012) | (0.017) | (0.016) |
| Wall is Bamboo | -0.056** | 0.047** | -0.008 |
| | (0.022) | (0.023) | (0.019) |

*(Continued)*

**Table 4.** (Continued)

| | (1) | (2) | (3) |
|---|---|---|---|
| | **Disposed in the Trash Can** | **Burned** | **Disposed in Land, Water, and Yard** |
| Floor is Not Soil | 0.067*** | 0.051** | -0.031* |
| | (0.024) | (0.021) | (0.017) |
| HH has Vehicle | -0.003 | 0.046*** | -0.042*** |
| | (0.008) | (0.011) | (0.010) |
| HH has Electronic Asset | -0.038** | 0.050*** | -0.008 |
| | (0.016) | (0.019) | (0.016) |
| HH has electricity | 0.019 | 0.041 | -0.002 |
| | (0.034) | (0.031) | (0.026) |
| Has no toilet | -0.074*** | -0.043*** | 0.061*** |
| | (0.017) | (0.016) | (0.014) |
| N | 9,626 | 9,626 | 9,626 |

*Statistical significance*: robust clustered standard error in parentheses with

* at 10 percent, ** at 5 percent, and *** at 1 percent level.

Notes: Coefficients are the marginal effects after probit regression. The samples used are post-matching observation or *common support only*, in which we exclude 35 observations from the control groups as the *off support*. Island dummies use Sumatra Island as the reference category.

the contaminated river, which is still used for daily needs or as infiltrated groundwater, caused 17 million cases of diarrhea in Indonesia, 39.6% of which were untreated [42].

Our empirical result in Table 4 confirms [35] that individuals who were living in rural areas, have low educational attainment, and come from a disadvantaged socioeconomic background, tend to not properly dispose of their solid waste in the trash can. From the supply side, the lack of funding causes the limited availability of waste facilities and services in Indonesia [50]; and the distance to final disposal sites from home is not a walking distance. Households do not want to be too bothered about taking the trip. In addition to that, the GoI has issued laws and regulations on waste management, such as Law No. 8 of 2008 on the Management of Municipal Solid Waste and Presidential Regulation No. 97 of 2017 on the Indonesian National Strategy Policy on Managing Domestic Waste and Domestic Waste Equivalents. However, the enforcement is still ineffective, easily caught and seen by everyday improper practices and violation of the rules in the neighborhood without penalties. The GoI, then, must provide the needed facilities and infrastructures to overcome these habits. The GoI must also enforce the regulation effectively.

## 5.4 Heterogeneous effect estimates

We subsample observations by demography and geography to observe whether our estimates' magnitudes are consistent or not across different sub-group of interest. Furthermore, the analysis identifies the priority target population for policymaking in reducing the ex-ante moral hazard by, for example, improving the waste management on the supply side. Overall, the government might target low-income groups, low education levels, and the population living in urban areas. The heterogeneous impacts are summarized in Table 5. The detailed results are as follows.

As for the outcome variable of disposing of waste to the trash can, lower-income households' sub-population consistently to have ex-ante moral hazards, even with higher magnitudes of effects. In addition, the negative effects are also found for female-headed households, low-educated, households living in Non-Jawa Island, and households living in urban areas and

**Table 5. Heterogenous analysis.**

| Subgroups | Waste Management Behavior | | | Obs. |
|---|---|---|---|---|
| | Disposing of waste in the Trash Can | Burned | Disposing of waste in Land, Water, and Yard | |
| Male | -0.024 | 0.038 | 0.006 | 7,892 |
| | (0.020) | (0.027) | (0.023) | |
| Female | -0.106*** | 0.042 | 0.014 | 1,734 |
| | (0.038) | (0.057) | (0.051) | |
| Education lower than elementary school | -0.064*** | 0.067 | -0.003 | 2,718 |
| | (0.020) | (0.042) | (0.040) | |
| Education equals to / higher than elementary school | -0.028 | 0.027 | 0.010 | 6,908 |
| | (0.025) | (0.031) | (0.026) | |
| Quartile 1 (poorest) | -0.071*** | 0.037 | 0.025 | 1,994 |
| | (0.023) | (0.045) | (0.043) | |
| Quartile 2 | -0.041 | 0.062 | -0.028 | 2,135 |
| | (0.031) | (0.049) | (0.047) | |
| Quartile 3 | 0.045 | -0.056 | 0.004 | 1,881 |
| | (0.042) | (0.057) | (0.049) | |
| Quartile 4 | -0.081 | 0.039 | 0.010 | 1,922 |
| | (0.063) | (0.074) | (0.059) | |
| Quartile 5 (richest) | -0.043 | 0.001 | 0.068 | 1,578 |
| | (0.086) | (0.110) | (0.068) | |
| 40% Lowest of Per Capita Expenditure | -0.056*** | 0.053 | 0.001 | 4,129 |
| | (0.019) | (0.033) | (0.031) | |
| 60% Highest of Per Capita Expenditure | -0.009 | -0.008 | 0.018 | 5,497 |
| | (0.033) | (0.040) | (0.032) | |
| Rural | -0.011 | 0.003 | 0.005 | 5,012 |
| | (0.015) | (0.035) | (0.033) | |
| Urban | -0.068** | 0.075** | 0.010 | 4,614 |
| | (0.031) | (0.033) | (0.025) | |
| Java Region | -0.028 | 0.063* | -0.012 | 5,137 |
| | (0.024) | (0.033) | (0.028) | |
| Non-Java Region | -0.052** | 0.015 | 0.023 | 4,489 |
| | (0.026) | (0.037) | (0.032) | |
| Community with Garbage Collector | -0.039** | 0.049* | 0.002 | 8,956 |
| | (0.019) | (0.025) | (0.022) | |
| Community without Garbage Collector | 0.000 | -0.043 | 0.039 | 670 |
| | (0.000) | (0.105) | (0.103) | |

*Statistical significance*: robust clustered standard error in parentheses with

* at 10 percent, ** at 5 percent, and *** at 1 percent level.

*Note*: Outcome variables are the dummy variable of waste management: (1) Disposed in the trash can; (2) Burned; (3) Disposed into river, yard, or pit. Some covariates in difference regression have been omitted for the convenience. Quartiles are derived from 2007 per capita expenditure. All regression results are based on balanced panel of 17,333 individuals for two periods, in which 17,301 individuals are lie on the region of common support after matching. Source: Authors' analysis based on Indonesia Family Life Survey (IFLS) 2007 & 2014 household-level panel data.

population living with the presence of waste collection system. The treatment group of these sub-populations has a smaller number of households that manage waste in the range of 5.6 to 7.1 percentage points. We found the consistent null effect for outcome variables of burning trash and throwing the trash to land or open dumping.

**Table 6. Coefficient stability test.**

|  | (1) | (2) | (3) |
|---|---|---|---|
|  | **Disposed in the Trash Can** | **Burned** | **Disposed in Land, Water, and Yard** |
| Subsidized Insurance x POST | -0.032* | 0.049* | -0.018 |
|  | (0.017) | (0.025) | (0.023) |
| Adjusted coefficients | -0.017 | 0.056 | -0.036 |
| R-squared | 0.446 | 0.147 | 0.127 |
| R-squared Maximum | 0.580 | 0.191 | 0.165 |
| Covariates | YES | YES | YES |
| N | 9,033 | 9,033 | 9,033 |

Note: The estimates use OLS (Ordinary Least Square) as the command *psacalc* to implement coefficient stability test based on Oster (2019) only available for linear estimation. *Statistical significance*: robust clustered standard error in parentheses with

* at 10 percent, ** at 5 percent, and *** at 1 percent level. Covariates used are the same as in Table 5.

## 5.5 Robustness check

Our estimates in Table 4 might not be completely free from biases. Therefore, we perform a coefficient stability test based on [46]. The results are summarized in Table 6. Overall, the adjusted coefficients are lower in absolute terms for disposing the waste to the trash can as the outcome variable (first-column) but higher for burning trash (second-column) and throwing the trash to land or open dumping (third-column). The adjusted coefficient in the first column makes the conclusion unchanged, it is still negative and statistically significant at ten percent level. There is an effect of subsidized insurance on the probability of households disposing of the waste to the trash can. It could be the case that the reduced number of households throwing the waste to trash can is shifted to burning trash. The inspection to the data reveals that there have been numbers of household shifting from throwing trash to land to become burning trash among the subsidized ones. It is about 176 households or about 39.64% of the total households practicing burning trash in 2014.

Moreover, the adjusted coefficient in the second column indicates the tentative finding that subsidized insurance increases the probability of households burning trash as the magnitude becomes larger. Last, the adjusted coefficient in the third column holds the case that subsidized insurance has no effect on the probability of households throwing the trash to land or open dumping. Based on the robustness test, our overall conclusion is unchanged.

## 6. Conclusion

This paper investigates the impact of a subsidized health insurance program on how lower-income households manage their domestic waste. We statistically find evidence of a projection of ex-ante moral hazard of fewer people to properly dispose of their waste into the trash can after the selected households obtained the program. On the external validity, we acknowledge the potential low base effect in which the point of estimates might be overstated compared to the recent situation as the share of the subsidized group becomes larger. Accordingly, we infer that our estimates serve as the upper bound of the effects. It indicates irresponsible behavior of the insured households toward their home environment, which would then imply their health condition.

The finding of this study will help the government not only focus on the curative action but more importantly on preventive ones by imposing stricter prohibitions for subsidized insurance recipients to keep their good behavior of disposing waste to trash can. However, the

government needs to find more effective approaches for behavior change initiatives. Incentives are one example of the initiatives and are considered economically attractive for lower-income households. Another example is to improve household awareness of the benefits of domestic waste management. Social campaigns and marketing through various media, such as electronic, mass, and social media, are possible ways to raise awareness. Through this social dissemination, social learning is expected to occur. Households are then expected to engage more actively in waste management. The government can also increase waste sorting and recycling training for low-income households, as this will provide them with knowledge on how to make objects from recycled materials. This creates new opportunities for income-generating activities. Furthermore, social empowerment, particularly for women, can hasten the implementation of domestic waste management, including the 3R program (reuse, reduce, and recycle), and, ultimately, realizing the circular economy (zero waste).

Due to limited lower-income household knowledge, limited waste disposal facilities, and not all households being capable of paying for private garbage collector services, households tend to dispose of their waste improperly. Households then must constantly be informed of the danger of improper waste disposal and littering as it will bring long-term impacts to their physical and environmental health. The government should also prioritize the provision of adequate waste disposal facilities across the country. The imminent reduced improper waste disposal will lead to a healthier household living environment which, in turn, will reduce the government's potential extra budget to mitigate health-related problems due to improper waste disposal and the pollution it may cause.

## Supporting information

**S1 Appendix.**
(DOCX)

## Acknowledgments

We thank Vid Adrison, Ph.D. and M.H. Yudhistira, Ph. D. for their constructive comments in improving the manuscript during the preliminary dissemination seminar at the Department of Economics, FEB UI.

We also thank the reviewers for their valuable input and suggestion in improving the manuscript during the review process for the publication.

## Author Contributions

**Conceptualization:** Beta Yulianita Gitaharie, Dwini Handayani.

**Data curation:** Rus'an Nasrudin, Lovina Aisha Malika Putri, Muhammad Abdul Rohman.

**Formal analysis:** Beta Yulianita Gitaharie, Rus'an Nasrudin, Ayu Putu Arantza Bonita.

**Funding acquisition:** Beta Yulianita Gitaharie.

**Methodology:** Rus'an Nasrudin.

**Project administration:** Ayu Putu Arantza Bonita.

**Software:** Lovina Aisha Malika Putri, Muhammad Abdul Rohman.

**Supervision:** Beta Yulianita Gitaharie.

**Visualization:** Lovina Aisha Malika Putri, Muhammad Abdul Rohman.

**Writing – original draft:** Beta Yulianita Gitaharie, Rus'an Nasrudin, Ayu Putu Arantza Bonita, Dwini Handayani.

**Writing – review & editing:** Beta Yulianita Gitaharie, Rus'an Nasrudin, Ayu Putu Arantza Bonita, Dwini Handayani.

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
