## [Decision Letter · Decision Letter 0]

24 Apr 2022

PONE-D-22-00455Is there an ex-ante moral hazard on Indonesia’s health insurance? An impact analysis on household waste management behaviorPLOS ONE

Dear Dr. Gitaharie,

Thank you for submitting your manuscript to PLOS ONE. After careful consideration, we feel that it has merit but does not fully meet PLOS ONE’s publication criteria as it currently stands. Therefore, we invite you to submit a revised version of the manuscript that addresses the points raised during the review process.

We look forward to receiving your revised manuscript.

Kind regards,

Kumaran Subramanian

Academic Editor

PLOS ONE

Journal Requirements:

3. We note that you have stated that you will provide repository information for your data at acceptance. Should your manuscript be accepted for publication, we will hold it until you provide the relevant accession numbers or DOIs necessary to access your data. If you wish to make changes to your Data Availability statement, please describe these changes in your cover letter and we will update your Data Availability statement to reflect the information you provide

Additional Editor Comments (if provided):

Dear authors

The authors should carry out the reviewers suggestions

Need to improve the quality of the manuscript

Check the typo graphical errors in the manuscript

The article needs major revision for further process

Reviewers' comments:

Reviewer's Responses to Questions

**Comments to the Author**

1. Is the manuscript technically sound, and do the data support the conclusions?

Reviewer #1: Yes

2. Has the statistical analysis been performed appropriately and rigorously? 

Reviewer #1: Yes

3. Have the authors made all data underlying the findings in their manuscript fully available?

Reviewer #1: Yes

4. Is the manuscript presented in an intelligible fashion and written in standard English?

Reviewer #1: Yes

5. Review Comments to the Author

Reviewer #1: Dear Authors,

The manuscripts is well-organized and well written. The topic is interesting and significant in the current situation. I would suggest few minor revisions to enhance the quality of your manuscript.

1. You have mentioned the data was collected from surveys in 2007 and 2014. A short explanation how these data still relevant for current situation in Indonesia would be useful.

2. I'm sure the findings of this study would be relevant for the Indonesian government. But, you need to be more precise on stating how the government can use this findings.

Thank you and good luck!

6. PLOS authors have the option to publish the peer review history of their article (what does this mean?). If published, this will include your full peer review and any attached files.

Reviewer #1: No

---

## [Author Response · Author response to Decision Letter 0]

17 Jun 2022

The followings are our responses to the referee’s comments. 

Comment #1

You have mentioned the data was collected from surveys in 2007 and 2014. A short explanation how these data still relevant for current situation in Indonesia would be useful.

Response: 

Thank you for raising this important concern. Accordingly, we provide the following responses. 

• IFLS is a longitudinal survey that have been conducted since 1993 by Rand Corporation with the latest available wave was from 2014 survey (https://www.rand.org/well-being/social-and-behavioral-policy/data/FLS/IFLS.html). The use of a longitudinal survey in our study is advantageous because as it is better in capturing the respondents’ changes both in the outcome and variable of interest as they are followed over time. Moreover, in our best of knowledge, there is no other potential data for such purpose. The socioeconomic national survey (SUSENAS) has some thematic survey in particular year containing health aspects, such as in 2017. In addition, the Demographic Health Survey/DHS (SDKI) and the Basic Health Research Survey (RISKESDAS) are potential but do not contain our variables of interest. Nevertheless, these datasets are cross-sectional so they are less ideal for our purpose. 

• The dynamic variation that we explore was coming from 2014 as the latest, we argue that the data still provide basis for internal validity for the particular time period. Moreover, the external validity of the estimate might be differ for the recent time period. As depicted in Table 2, there has been tendency of increasing share of subsidized insurance group toward recent time. Accordingly, we can expect that the low base effect could drive a larger effect in our estimate compared to scenario when we use the most recent data. 

• Accordingly, we add a sentence to briefly explain why we use 2014 as the end line period in lines 68-70;

“We exploit the variation of the household’s health insurance between the baseline period of 2007 and the end line period of 2014 as the most suitable recent data that is available for analysis.2 

---- 

2 To the best of our knowledge, there is no other potential data for such purpose. The socioeconomic national survey (SUSENAS) has some thematic surveys in a particular year containing health aspects, such as in 2017. In addition, the Demographic Health Survey/DHS (SDKI) and the Basic Health Research Survey (RISKESDAS) are potential but do not contain our variables of interest. Nevertheless, these datasets are cross-sectional so they are less ideal for our purpose.”

• and qualify its external validity toward the recent time context in lines 479-482:

“On the external validity, we acknowledge the potential low base effect in which the point of estimates might be overstated compared to the recent situation as the share of the subsidized group becomes larger. Accordingly, we infer that our estimates serve as the upper bound of the effects.”

• Moreover, the data used in our study are also still relevant as the dynamics of outcome and key variables of interest have not changed much since 2014 (see Table 2). 

• We also added the percentage of insurance beneficiary households in Panel B Table 2. 

Comment #2

I'm sure the findings of this study would be relevant for the Indonesian government. But, you need to be more precise in stating how the government can use these findings.

Response:

Thank you to referee for this important suggestion. Accordingly, we provided the following revisions:

• We added the following sentences in the Introduction (lines 90-92): 

“The finding of this study is expected to help the Indonesian government to design a better intervention concerning adverse side effect of health insurance subsidy.”

• Also, we revised the following sentences in the Conclusion (lines 483-491):

“The finding of this study will help the government not only focus on the curative action but also on preventive ones by imposing stricter prohibitions for subsidized insurance recipients to not burn their domestic waste and manage them properly. However, the Indonesian government needs to train the lower-income group to manage waste sorting and recycling. It is also likely the training result gives the lower-income group opportunity for income-generating activities. The reduced waste burning will lead to a healthier household living environment which, in turn, will reduce the potential government extra budget to mitigate health-related problems due to waste burning air pollution.”

---

## [Decision Letter · Decision Letter 1]

25 Jul 2022

PONE-D-22-00455R1Is there an ex-ante moral hazard on Indonesia’s health insurance? An impact analysis on household waste management behaviorPLOS ONE

Dear Dr. Gitaharie,

Thank you for submitting your manuscript to PLOS ONE. After careful consideration, we feel that it has merit but does not fully meet PLOS ONE’s publication criteria as it currently stands. Therefore, we invite you to submit a revised version of the manuscript that addresses the points raised during the review process.

The second reviewer has confirmed the high quality of the paper, however the first reviewer has some further minor comments which shouldn't be too problematic to address.

We look forward to receiving your revised manuscript.

Kind regards,

Alison Parker

Academic Editor

PLOS ONE

Journal Requirements:

Reviewers' comments:

Reviewer's Responses to Questions

**Comments to the Author**

1. If the authors have adequately addressed your comments raised in a previous round of review and you feel that this manuscript is now acceptable for publication, you may indicate that here to bypass the “Comments to the Author” section, enter your conflict of interest statement in the “Confidential to Editor” section, and submit your "Accept" recommendation.

Reviewer #1: (No Response)

Reviewer #2: All comments have been addressed

2. Is the manuscript technically sound, and do the data support the conclusions?

Reviewer #1: Partly

Reviewer #2: Yes

3. Has the statistical analysis been performed appropriately and rigorously? 

Reviewer #1: I Don't Know

Reviewer #2: Yes

4. Have the authors made all data underlying the findings in their manuscript fully available?

Reviewer #1: Yes

Reviewer #2: Yes

5. Is the manuscript presented in an intelligible fashion and written in standard English?

Reviewer #1: Yes

Reviewer #2: Yes

6. Review Comments to the Author

Reviewer #1: The study has been well written, so does the introduction section. It provides clear argument about the importance of the study. However, I would like to recommend one small thing for this section. The authors have mention that they have found an evidence of ex-ante moral hazard in the subsidized health insurance recipients on how they manage their household waste. The insured households are more likely to burn their domestic waste than the uninsured. How do we know that this act of burning was a result of subsidizing health insurance? was it more related to a common behaviour of group of people instead?

Another explanation about how do you relate the health insurance to a waste management would also be useful for this section.

In the the conclusion, the authors offer recommendation for Indonesian government that they need to advise the insured families not to burn their waste. Is it the advise you really want to take out from your study findings? I believe the authors are able to provide some more ideas to their government about waste management that they believe it relate to the provision of health insurance.

Reviewer #2: The manuscript has been revised thoroughly, the data support the conclusions, the statistical analysis has been performed appropriately and rigorously, the authors have made all data underlying the findings in their manuscript fully available, the manuscript is presented in an intelligible fashion and written in standard English.

7. PLOS authors have the option to publish the peer review history of their article (what does this mean?). If published, this will include your full peer review and any attached files.

Reviewer #1: No

Reviewer #2: No

---

## [Author Response · Author response to Decision Letter 1]

18 Sep 2022

A point-by-point response to the reviewer are provided in the attached file (Response to Reviewers_Revision2.docx)

---

## [Decision Letter · Decision Letter 2]

10 Oct 2022

Is there an ex-ante moral hazard on Indonesia’s health insurance? An impact analysis on household waste management behavior

PONE-D-22-00455R2

Dear Dr. Gitaharie,

We’re pleased to inform you that your manuscript has been judged scientifically suitable for publication and will be formally accepted for publication once it meets all outstanding technical requirements.

Kind regards,

Alison Parker

Academic Editor

PLOS ONE

Additional Editor Comments (optional):

Reviewers' comments:

Reviewer's Responses to Questions

**Comments to the Author**

1. If the authors have adequately addressed your comments raised in a previous round of review and you feel that this manuscript is now acceptable for publication, you may indicate that here to bypass the “Comments to the Author” section, enter your conflict of interest statement in the “Confidential to Editor” section, and submit your "Accept" recommendation.

Reviewer #2: All comments have been addressed

2. Is the manuscript technically sound, and do the data support the conclusions?

Reviewer #2: Yes

3. Has the statistical analysis been performed appropriately and rigorously? 

Reviewer #2: Yes

4. Have the authors made all data underlying the findings in their manuscript fully available?

Reviewer #2: Yes

5. Is the manuscript presented in an intelligible fashion and written in standard English?

Reviewer #2: Yes

6. Review Comments to the Author

Reviewer #2: The revised manuscript version is well improved. Wish the findings of this manuscript can elevate well-waste-management in Indonesia.

7. PLOS authors have the option to publish the peer review history of their article (what does this mean?). If published, this will include your full peer review and any attached files.

Reviewer #2: No

---

## [Editor Report · Acceptance letter]

5 Dec 2022

PONE-D-22-00455R2 

Is there an ex-ante moral hazard on Indonesia’s health insurance? An impact analysis on household waste management behavior 

Dear Dr. Gitaharie:

I'm pleased to inform you that your manuscript has been deemed suitable for publication in PLOS ONE. Congratulations! Your manuscript is now with our production department. 

Kind regards, 

on behalf of

Dr. Alison Parker 

Academic Editor

PLOS ONE